# Characterization of Different Salt Forms of Chitooligosaccharides and Their Effects on Nitric Oxide Secretion by Macrophages

**DOI:** 10.3390/molecules26092563

**Published:** 2021-04-28

**Authors:** Ronge Xing, Chaojie Xu, Kun Gao, Haoyue Yang, Yongliang Liu, Zhaoqian Fan, Song Liu, Yukun Qin, Huahua Yu, Pengcheng Li

**Affiliations:** 1CAS and Shandong Province Key Laboratory of Experimental Marine Biology, Center for Ocean Mega-Science, Institute of Oceanology, Chinese Academy of Sciences, No. 7 Nanhai Road, Qingdao 266071, China; xuchaojie-qdio@foxmail.com (C.X.); kung1993@163.com (K.G.); yanghaoyue@qdio.ac.cn (H.Y.); fanzhaoqian@qdio.ac.cn (Z.F.); sliu@qdio.ac.cn (S.L.); ykqin@qdio.ac.cn (Y.Q.); yuhuahua@qdio.ac.cn (H.Y.); pcli@qdio.ac.cn (P.L.); 2Laboratory for Marine Drugs and Bioproducts, Pilot National Laboratory for Marine Science and Technology (Qingdao), No. 1 Wenhai Road, Qingdao 266237, China; lyliocas@163.com; 3University of Chinese Academy of Sciences, Beijing 100049, China

**Keywords:** pure chitooligosaccharides, chitooligosaccharides salts, characterization, promoting NO production

## Abstract

In this paper, chitooligosaccharides in different salt forms, such as chitooligosaccharide lactate, citrate, adipate, etc., were prepared by the microwave method. They were characterized by SEM, FTIR, NMR, etc., and the nitric oxide (NO) expression was determined in RAW 264.7 cells. The results showed that pure chitooligosaccharide was an irregular spherical shape with rough surface, and its different salt type products are amorphous solid with different honeycomb sizes. In addition to the characteristic absorption peaks of chitooligosaccharides, in FTIR, the characteristic absorption of carboxyl group, methylene group, and aromatic group in corresponding acid appeared. The characteristic absorption peaks of carbon in carboxyl group, hydrogen and carbon in methyl, methylene group, and aromatic group in corresponding acid also appeared in NMR. Therefore, the sugar ring structure and linking mode of chitooligosaccharides did not change after salt formation of chitooligosaccharides. Different salt chitooligosaccharides are completely different in promoting NO secretion by macrophages, and pure chitooligosaccharides are the best.

## 1. Introduction

Chitosan is the deacetylation product of chitin from the shell of crustacean, plant, and fungal cell walls. It is the only natural polysaccharide with cation found in nature so far. At the same time, chitosan has been widely used in medicine [1,2], agriculture [3,4], food [5,6], papermaking [7,8], wastewater treatment [9,10], and other fields due to its natural non-toxic, biodegradable, and unique physical and chemical structure, and has become a hot spot in the research of new biological materials recently. However, because chitosan is insoluble in water, alkaline solution, and organic solvent, and is only soluble in some organic acids and dilute inorganic acid, such as acetic acid, dilute hydrochloric acid, etc., its application is greatly limited. Therefore, the improvement of the solubility of chitosan has been one of the concerns of researchers at home and abroad. Some studies have shown that chitosan salt formed with organic and inorganic acids is soluble in water and has been widely studied as a polymer drug carrier and a novel drug absorption accelerator across nasal and intestinal epithelium [11,12]. However, different chitosan salt forms showed different drug release profiles. For example, chitosan glutamate salt is preferred for nasal drug delivery [13], however, chitosan hydrochloride salt showed a high intestinal absorption-enhancing effect [14]. In other words, in addition to molecular weight and degree of acetylation, salt type is also a major factor affecting the activity of chitosan.

Chitooligosaccharide, with the degree of polymerization less than 20 and an average molecular weight (MW) less than 3.9 kDa, is the degradation product of chitosan via enzymatic or chemical hydrolysis. Due to its low MW and good solubility in acid, neutral, and alkaline conditions, it has attracted the interest of many researchers for its versatility in agriculture, biomedical, and pharmaceutical applications, etc. The preparation of chitooligosaccharide is mainly a homogeneous reaction. In other words, chitosan is dissolved in acid solution, and then chitooligosaccharide is prepared by chemical or enzymatic methods. In most products and activity studies of chitooligosaccharide, generally speaking, chitosan is dissolved in acetic acid, lactic acid, or hydrochloric acid to prepare chitooligosaccharide acetate, chitooligosaccharide hydrochloride, and chitooligosaccharide lactate. Moreover, in terms of the three salt forms, the taste and activity are also different due to the different salt forms formed by different acids. Cho et al. [15] found that chitooligosaccharide lactate significantly decreased the severity of fatigue compared with chitooligosaccharide hydrochloride, which showed no statistically significant anti-fatigue effects. This can be explained by the different ionization degree of different salt forms. According to the theory of acid–base distribution, chitooligosaccharide lactate is a kind of weak acid, most of which exists in the form of non-ionization under physiological pH value, while chitooligosaccharide hydrochloride is a strong acid, and preferentially exists in the form of ionization. As a result, their activities are quite different. In fact, chitosan dissolved in different acids can be degraded into corresponding salt form chitooligosaccharides. For example, Hai et al. [16] dissolved chitosan in lactic acid and degraded chitosan with hydrogen peroxide to obtain chitooligosaccharide lactate. ^1^HNMR showed that the product of chitooligosaccharide lactate prepared by Hai et al. has one more hydrogen absorption peak in the methyl group than chitosan. Li et al. [17] dissolved chitosan in glacial acetic acid and prepared chitooligosaccharide acetate by enzymatic hydrolysis of chitosan. ^13^CNMR showed that a new peak appeared at about 22 ppm, which could be explained as the absorption peak of the methyl group in chitooligosaccharide acetate. Qin et al. [18] dissolved chitosan in hydrochloric acid and obtained chitooligosaccharide hydrochloride by oxidative degradation of chitosan. According to ^13^CNMR, its structure is consistent with that of chitosan. Some studies showed the potential of COS as an immunostimulatory agent. Zhang et al. [19] showed that chitooligosaccharide exerts this immuno-stimulating activity by interacting with membrane receptors on the macrophage surface. Zheng et al. [20] found that chitooligosaccharide has promoted the expression of the genes of vital molecules in NF-κB and AP-1 pathways and induced the phosphorylation of protein in RAW 264.7 macrophage. All the raw materials they used were chitooligosaccharide acetate. In addition to chitooligosaccharide acetate, chitooligosaccharide hydrochloride, and chitooligosaccharide lactate, other salt form chitooligosaccharides have not been reported. Therefore, in this paper, 10 kinds of salt form chitooligosaccharides were prepared. Physicochemical properties of different salt form chitooligosaccharides such as morphology, chemical structure, water solubility, thermal behavior, etc., were characterized compared with salt-free chitooligosaccharides. Moreover, the immune activity of chitooligosaccharides with different salt forms was preliminarily studied.

## 2. Results and Discussion

### 2.1. Morphology of Chitooligosaccharide and Its Salts

The morphology of chitooligosaccharide and its salts was observed by SEM, as shown in Figure 1. Figure 1 showed that there was an obvious different morphology between chitooligosaccharide and its salts. Chitooligosaccharide was of irregular spherical shape with rough surface, however, in addition to chitooligosaccharide citrate (COS-Cit), chitooligosaccharide malate (COS-Mal), and chitooligosaccharide succinate (COS-Suc), other different salt products are massive, with different honeycomb sizes. COS-Cit, COS-Mal, and COS-Suc are all massive, so their structures are similar. For other salt types, although the size of the honeycomb is different, the morphology and structure of different salt forms of chitooligosaccharides are basically the same. Different morphology of chitooligosaccharide and its salts showed that the salts had higher allomorphic nature than the pure chitooligosaccharides.

### 2.2. FTIR Analysis of Chitooligosaccharide and Its Salts

The infrared spectrum (FTIR) analysis of pure chitooligosaccharides (COS) and chitooligosaccharides with different salt types is shown in Figure 2. The FTIR spectra of COS display characteristic absorption bands at 3200–3400 cm^−1^ and 2874 cm^−1^, which represent the presence of NH_2_ group, OH group, and CH_2_ group. The absorption peaks at 1589 and 1375 cm^−1^ are the amide II band and the amide III band. The special absorption peak of 1158–890 cm^−1^ corresponds to the structure of polysaccharide and the 1,4-glycoside bonds in the COS structure [21,22,23]. It can be seen from Figure 2 that compared with COS, most of the absorption bands of COS are retained by chitooligosaccharides of different salt types, especially the special absorption peaks representing the structure of polysaccharides, which shows that the structure of chitooligosaccharides of different salt types is still consistent with COS. In the spectrum of chitooligosaccharide hydrochloride (COS-HCl), 1589 cm^−1^ in the original COS spectrum disappeared, resulting in two new peaks of 1614 and 1506 cm^−1^, indicating the electrostatic interaction between the amino group in COS and hydrochloric acid [24]. For chitooligosaccharide acetate (COS-Ace), the amide II band at 1538 cm^−1^ shifted to a low wave number, which showed that the amino group is protonated to form a carboxylate between the –COO^−^ group of the acid and the NH^3+^ group of COS, and the free amino group numbers decreased [25]. For other salt types of chitooligosaccharide: chitooligosaccharide lactate (COS-Lat), chitooligosaccharide citrate (COS-Cit), chitooligosaccharide malate (COS-Mal), chitooligosaccharide tartrate (COS-Tar), chitooligosaccharide acetylsalicylate (COS-Sal), chitooligosaccharide succinate (COS-Suc), chitooligosaccharide adipate (COS-Adi), and chitooligosaccharide glutamate (COS-Glu), the absorption peak of 1589 cm^−1^ of the amide II band in the COS spectra shifted to the low wave number because they were carboxylates like COS-Ace. The results of the infrared spectrum of COS-Lat show some characteristic absorption peaks of lactate, for example, 1720 cm^−1^ is the C=O stretching vibration peak, 1463 cm^−1^ is the -CH_3_ bending vibration peak, 823 cm^−1^ is the C-C stretching vibration peak, 1372 cm^−1^ is the -CH_3_ symmetric vibration absorption peak, and 1222 cm^−1^ is the -C=O bending vibration absorption peak, indicating that there is still some lactate in COS-Lat. However, the existence of the characteristic peaks of 1567 cm^−1^ amide II band and 1158–890 cm^−1^ sugar structure in the chitooligosaccharide indicated that COS-Lat was successfully prepared. For COS-Cit, COS-Mal, and COS-Tar, there are several carboxylic acid groups in their molecules. From the infrared spectrum, it can be seen that there are about 1710 cm^−1^ stretching vibration absorption peaks of carbonyl C=O, indicating that only part of the carboxylic acid groups in the three kinds of acid molecules are linked to the amino groups in the chitooligosaccharide, that is, the hydrogen on some carboxylic acid groups is not dissociated. This also shows that the structural formula of the reaction product drawn in this paper is correct. For the COS-Sal, due to the introduction of acetylsalicylic acid, the characteristic absorption peak of benzene ring appeared in the infrared spectrum, such as 1604 and 1483 cm^−1^ benzene ring skeleton vibration peak, and 757 cm^−1^ indicates that acetylsalicylic acid is ortho-substitution. In addition, 1740 cm^−1^ carbonyl C=O in acetylsalicylic acid disappeared, and a new 1656 cm^−1^ carbonyl peak appeared in COS-Sal, which proved that the carboxylic acid group of acetylsalicylic acid was linked to the amino group in chitooligosaccharide. For COS-Suc and COS-Adi, the existence of the antisymmetric COO^−^ characteristic peak at 1690 cm^−1^ indicates the existence of carboxylate ions. In addition, due to the different length of the carbon chain between succinic acid and adipic acid, there is a plane swing characteristic absorption peak of methylene -(CH_2_)- at 801 cm^−1^ in COS-Suc and and 734 cm^−1^ in COS-Adi. In the FTIR of COS-Glu, the absorption band at 1537 cm^−1^ was characteristic of the amino deformation mode of glutamate and amide II band of COS-Glu, suggesting the formation of chitooligosaccharide glutamate.

### 2.3. ^1^HNMR and ^13^CNMR Spectroscopy Analysis of Chitooligosaccharide and Its Salts

The chemical structure of chitooligosaccharide and different salt types were further investigated by both ^1^HNMR and ^13^CNMR spectroscopy in D_2_O or CD_3_COOD-D_2_O, and the results are shown in Figure 3. As shown in Figure 3, pure COS exhibited a typical signal peak at δ 1.91 ppm for ^1^HNMR spectra and a weak signal at δ 22.22 ppm for ^13^CNMR spectra, corresponding to methyl protons of the N-acetyl groups, which confirmed the existence of GlcNAc residue. For ^1^HNMR spectra of pure COS, the signals at 2.62 and 4.42 ppm were attributed to H-2 and H-1 respectively, and multiple peaks at 3.26–3.79 ppm were assigned to H-3, H-4, H-5, and H-6. Similar to FTIR analysis, the ^1^HNMR spectrum of pure COS is similar to that of the original chitosan, which coincided with the reported data [26,27]. For the ^13^CNMR spectrum of pure COS (Figure 3b), the strong signals at 56.49, 60.07, 69.51, 74.76, 76.10, and 101.88 ppm were attributed to C2, C6, C3, C5, C4, and C1, respectively. Compared with original chitosan [28], there was also almost no change in the COS spectrum. There was no obvious carbonyl group signal around 175 ppm, which showed that there was no ring opening impurity. In the comparison of the spectra of pure COS and its different salt types’ samples, there were similar chemical shifts for both ^1^HNMR (H1–H6, NHCOCH_3_) and ^13^CNMR (C1–C6 and NHCOCH_3_). At the same time, the corresponding characteristic absorption peaks of acid appeared in the ^1^HNMR and ^13^CNMR spectra of different salt type chitooligosaccharides. In the ^1^HNMR spectra, the signal peaks of COS-HCl and COS are consistent. For COS-Ace, COS-Lat, COS-Sal, and so on, due to the presence of methyl, methylene, and aromatic rings in the acid, corresponding signal peaks appear in the ^1^HNMR spectra. For COS-Ace, the strong signal at 1.79 ppm was attributed to methyl protons of acetic acid. In COS-Lat, the strong signals at 1.28 and 1.20 ppm were assigned to H7 and H8, which were signals of hydrogen protons on tertiary carbon (CH) and primary carbon (CH_3_) in lactic acid. In COS-Cit, two new signal peaks, 2.74 (H7) and 2.64 ppm (H8), are mainly the signals of hydrogen protons on the secondary carbon (CH_2_) in citric acid. For COS-Mal, new signals at 2.51–2.70 ppm (H7) were assigned to methylene (CH_2_) in malic acid. Because of the influence of hydroxyl group, the new signal peak appeared at 4.27 ppm, which was the signal peak of hydrogen in methine (CH) in malic acid. For COS-Tar, due to the presence of two methine groups (CH) affected by hydroxyl groups in tartaric acid, a signal peak of hydrogen in methine was observed at 4.44 ppm. COS-Sal shows the aromatic hydrogen at 6.79–7.66 ppm. In COS-Suc, the signal peaks at 2.42–2.53 ppm were corresponding to H7 and H8, which were the methylene groups (CH_2_) in succinic acid. However, for COS-Adi, due to the influence of carboxyl group, two methylene (CH_2_) appeared at 2.06–2.16 ppm (H7 and H10), and two methylene (CH_2_) appeared at about 1.44 ppm (H8 and H9). For COS-Glu, the signal peaks at 2.22–2.32 ppm were attributed to H7 and H9, which were the methine (CH) and methylene (CH_2_) groups affected by the carboxyl group and amino group in glutamic acid. Another new signal peak at 1.96 ppm was corresponding to H8 (CH_2_). In addition, the new peak with a chemical shift of 8.2 ppm in the ^1^HNMR spectrum is generally the hydrogen of the carboxyl group not involved in the reaction in the acid. The results of the ^1^HNMR spectra showed that different acid forms of chitooligosaccharides were successfully prepared. Furthermore, for ^13^CNMR, the chemical shifts and new absorption peaks of different salt type COS are consistent with the results of ^1^HNMR. In addition to COS-HCl, other salt type chitooligosaccharides all contain carboxyl group. According to the position and amount of carboxyl groups, there are new absorption peaks of carboxyl group in 174–181 ppm of the ^13^CNMR spectrum. The carbon absorption peaks of primary, secondary, tertiary, quaternary, and aromatic carbon in different acids also appeared in the related salt type chitooligosaccharides. They are all marked in Figure 3b. For example, in the spectrum of COS-Ace, the additional resonances at 22.56 ppm assigned to CH_3_ carbon representing acetate functionality were observed. For COS-Sal, aromatic carbon of the acetylsalicylic acid at 116.35–134.37 ppm appeared. Therefore, these results confirmed the successful link of different acids and COS. Moreover, it was further confirmed by NMR spectra that the chemical monomeric structures of chitosan remained for different salt type chitooligosaccharides.

### 2.4. XRD Analysis of Chitooligosaccharide and Its Salts

Figure 4 shows the X-ray diffraction patterns of the pure COS and its different salt form samples ranging from 5 to 80°. The X-ray diffraction pattern of the initial chitosan has two characteristic peaks at 2θ = 10.4° and 20.0° [18,28,29,30]. Compared with the original chitosan, COS becomes amorphous due to the decrease of crystallinity, and only one absorption peak appeared at 2θ = 22.4° [28,31]. As shown in Figure 4, the main characteristic peaks of different salt type chitooligosaccharides are about 2θ = 22.4°, which is consistent with the main characteristic absorption peak of pure COS. However, some salt type chitooligosaccharides have characteristic absorption peaks of related acid types due to the influence of acid types. In the case of COS-HCl, the peak at 2θ = 23.14° became low and broad, which indicated that the crystallinity of the chitooligosaccharide salt had been further decreased. This might be explained by the fact that the formation of –NH_3_^+^Cl^−^ caused the deformation of the hydrogen bonds in COS molecules [32]. However, for COS-Ace, COS-Lat, COS-Cit, COS-Mal, COS-Tar, and COS-Glu, the intensity of the peak of each sample increased compared with pure COS. This might be because the carboxyl group and hydroxyl group contained in the acid form hydrogen bonds with some hydroxyl groups in the chitooligosaccharide after the introduction of different acid forms into the salt, thus increasing the crystallinity of these salt type chitooligosaccharides. Furthermore, in the case of COS-Sal, COS-Suc, and COS-Adi, because of the introduction of acetylsalicylic acid, succinic acid, and adipic acid in the salt form of chitooligosaccharides, their crystal structures were also reflected in XRD. For example, the new peaks of 10°, 17°, 25°, 28°, and 30° of COS-Sal were speculated as the aromatic ring structure of acetylsalicylic acid. Due to the similar structure of succinic acid and adipic acid, COS-Suc and COS-Adi showed new peaks at 26° and 31°. However, the other peaks in COS-Sal, COS-Suc, and COS-Adi might represent the packing structure induced from a pi–pi stacking or aromatic–aromatic interaction of Phe [33].

### 2.5. Thermal Behavior Analysis of Chitooligosaccharide and Its Salts

Figure 5 depicts the TGA, DTG, and DSC curves of pure COS and its different salt types. DSC and TGA were selected to characterize the thermal properties and DTG was used to analyze the thermal decomposition behavior of pure COS and its different salt types. As shown in Figure 5, both COS and COS-HCl showed two stages in weight loss, COS-Suc and COS-Glu have four weight loss stages, while there are three weight loss stages in chitooligosaccharides of other salt forms. The first stage of weight loss of all samples is the loss of water adsorbed in the samples. Due to the different water-holding capacity of each sample, the maximum weight loss temperature of water loss is different. It can be seen from Figure 5 that COS-C has the strongest water-holding capacity, with the maximum weight loss temperature of 98 °C, followed by COS-Tar. The weaker water-holding capacity is of COS-Lat, COS-Sal, COS-Suc, and COS-Adi, and the maximum weight loss temperature is 76–78 °C. Chitooligosaccharides of other salt types have similar water-holding capacity with COS, and the maximum weight loss temperature is about 83 °C. The second stage of weight loss of COS is due to the cracking and breaking of chitooligosaccharide, which produces small molecular volatile products. The maximum weight loss temperature is 283 °C. The second stage of weight loss of COS-HCl is significantly higher than that of COS, which may be due to the formation of chitooligosaccharide and volatile HCl gas at high temperature, and then the chitooligosaccharide further breaks to produce small molecular volatile products, and the maximum weight loss temperature is 224 °C. For COS-Ace, COS-Lat, COS-Cit, COS-Mal, COS-Tar, COS-Sal, and COS-Adi, the second stage of weight loss is due to the decomposition of the carboxyl group in acid or C-N bond breaking in amide bond. Due to the different structure and content of the carboxyl group in each acid, their weight loss and maximum weight loss temperature are quite different. For example, citric acid, malic acid, and tartaric acid contain two or more carboxyl groups, in which methylene is used to connect carboxyl and hydroxyl groups. The structure is basically similar. The maximum weight loss temperature of acetic acid and lactic acid is about 236 °C. The maximum weight loss temperature of acetic acid and lactic acid is 157 and 187 °C, respectively. Adipic acid is composed of four methylene groups and two carboxyl groups. The maximum weight loss temperature is 168 °C. The structure of acetylsalicylic acid is two carboxyl groups in the ortho position of the benzene ring, and the maximum weight loss temperature is 202 °C. The third stage of weight loss is due to the cleavage and fracture of chitooligosaccharides. However, due to the different acid types, the maximum weight loss temperature is also different. Similar to the effect of acid type in the second stage, the maximum weight loss temperature of chitooligosaccharide citrate and chitooligosaccharide malate is the highest, while the maximum weight loss temperature of chitooligosaccharide acetate and chitooligosaccharide adipate is lower. For COS-Suc and COS-Glu, there are four stages of weightlessness due to the influence of succinic acid and glutamic acid. The second stage is mainly the decomposition of succinic acid and glutamic acid, while the third and fourth stage mainly include the continuous decomposition of succinic acid and glutamic acid accompanied by the cleavage of chitooligosaccharide. The successful link of different acids and COS was further confirmed by thermal behavior.

### 2.6. Cell Viability

The MTT assay was used to detect the cytocompatibility of different concentrations of pure chitooligosaccharides and different salt type chitooligosaccharides in RAW 264.7 cells. The survival rate of the blank control group was 100%. As shown in Figure 6, the results showed that all kinds of chitooligosaccharides had good biocompatibility to RAW 264.7 cells after 24 h of treatment. Moreover, the cell viability with 50‒500 μg/mL pure chitooligosaccharides and different salt type chitooligosaccharides exceeded 80%, indicating that pure chitooligosaccharides and different salt type chitooligosaccharides had no obvious toxicity in RAW 264.7 cells. Furthermore, Figure 6 shows that with the increase of concentration, pure chitooligosaccharide increased the survival rate of cells, indicating that pure chitooligosaccharide has the effect of promoting growth. Chitooligosaccharides with different salt forms also had a growth-promoting effect, but the effect was not as good as chitooligosaccharides.

### 2.7. The Effects of Chitooligosaccharide and Its Salts on Nitric Oxide (NO) Secretion

Macrophages, as a bridge between innate immunity and acquired immunity, play an important role in the immune system. RAW 264.7, a mouse macrophage, has been widely used as a model for measuring cellular immune activity and its mechanism in vitro [34,35,36]. Activated macrophages produce nitric oxide (NO) and a variety of cytokines [37,38]. NO, a messenger molecule with a very short half-life, plays an important role in innate and acquired immune responses [39]. NO is closely related to immune system, cardiovascular disease, nerve, gastrointestinal, nephropathy, urinary, tumor, liver disease, diabetes, infection, trauma, and organ transplantation rejection. It is highly valued by researchers. To determine whether the immunostimulatory activities of chitooligosaccharides were related to its salt form, pure chitooligosaccharides and different salt type chitooligosaccharides were prepared, and their NO-promoting activities were measured using Griess reagent and the activity between them was compared. Therefore, we determined the NO expression in RAW 264.7 cells treated with 500 μg/mL pure chitooligosaccharides and different salt type chitooligosaccharides, as shown in Figure 7. The results indicated that it is completely different that chitooligosaccharides with different salt forms promote the secretion of NO by macrophages, and pure chitooligosaccharides are more active at promoting NO production than different salt type chitooligosaccharides. Among the chitooligosaccharides with different salt forms, COS-HCl was the best, followed by COS-Ace, and COS-Sal was the weakest. A possible reason may be that the content of glucosamine of pure chitooligosaccharides was higher than that of different salt type chitooligosaccharides. However, the specific reasons need to be further studied.

## 3. Materials and Methods

### 3.1. Materials

Pure chitooligosaccharides was bought from Zhejiang Jinke Pharmaceutical Co., Ltd. (Yuhuan, China), whose molecular weight (MW) is 1859 Da and degree of deacetylation (DD) is 90%. Chitosan from shrimp and crab shells, whose DD and MW were 86% and 658 × 10^3^ Da respectively, was purchased from Qingdao Yunzhou Biochemical Corp. (Qingdao, China). Hydrochloric acid, acetic acid, citric acid, malic acid, glutamic acid, tartaric acid, adipic acid, succinic acid, acetylsalicylic acid, and lactic acid, hydrogen peroxide (30% concentration), and other chemicals were of analytical grade. The Roswell Park Memorial Institute (RPMI) medium 1640 and antibiotics were provided by Gibco BRL. (Life Technologies, Shanghai, China). Fetal bovine serum (FBS) was purchased from HyClone (Thermo Fisher Scientific, Logan, UT, USA). 1-(4,5-dimethylthiazol-2-yl)-3,5-diphenyltetrazolium bromide (MTT) and LPS (Escherichia coli 0111: B4) were obtained from Sigma (St. Louis, MO, USA).

### 3.2. Methods

#### 3.2.1. Preparation of Chitooligosaccharides with Different Salt Forms

The chitosan powder (5 g) was introduced in an Erlenmeyer flask containing 100 mL acid solution of different concentrations (Table 1). Then, hydrogen peroxide (H_2_O_2_, whose final concentration is 1%) was added to the chitosan aqueous solution. The mixture solution was heated for degradation of chitosan by microwave irradiation. The degradation with microwave irradiation was performed in a laboratory microwave synthesis/extraction reaction station (Type: MAS-II) that was purchased from Shanghai SINEO Microwave Chemistry Technology Co., Ltd. (Shanghai, China). with a magnetic stirrer and an infrared reaction thermometer. The experiments were employed with the power of 900 W at 90 °C at 30 min. After reaction, 1 mL of the reaction mixture was taken out for gel permeation chromatography analysis. Then, the reaction mixture was cooled to room temperature (25 °C). Subsequently, the mixture was concentrated and lyophilized to different salt form chitooligosaccharides (see Figure 8). The deacetylation degree of the products was 91–93% by UV.

#### 3.2.2. Characterization Methods

Molecular weights of chitooligosaccharide and its salts were determined by Agilent 1260 gel permeation chromatography (Agilent Technologies Inc., Santa Clara CA, USA) equipped with a refractive index detector. TSK G3000-PWXL columns were used. Mobile phase was 0.2 M CH_3_COOH/0.1 M CH_3_COONa at a flow rate of 1.0 mL/min, with column temperature at 30 °C. The concentration of all samples was 0.4% (*w*/*v*). Dextrans with molecular weight (MW) 36,800, 13,050, 9750, 5250, 2700, and 180 Da (National Institutes for Food and Drug Control, China) were standards which were used to calibrate the column.

Morphology of chitooligosaccharide and its salts was observed by a scanning electron microscope (MX 2000, Cam Scan, Cambridge, England). Scanning electron photomicrography of the gold-coated sample was taken at appropriate magnification.

Fourier transform infrared (FTIR) spectrum of chitooligosaccharide and its salts was measured in the 4000–400 cm^−1^ regions using a Thermo Scientific Nicolet iS10 FTIR spectrometer (Thermo Fisher Scientific Inc., Waltham, Ma, USA) and KBr discs.

^1^HNMR and ^13^CNMR spectra were recorded on a 500 MHz Agilent DD2 500 spectrometer for structural analysis in chitooligosaccharide and its salts. Measurements were recorded at 25 °C. Samples were dissolved at D_2_O (10 mg/mL) and pulse sequence was s2pul, relaxation delay was 1.0000 s. Acquisition time was 2.0447 s with 16 scans and spectral width was 8012.8 Hz for ^1^HNMR. Acquisition time was 1.0486 s and spectral width was 31,250 Hz for ^13^CNMR. For ^13^CNMR, number of scans of different samples are different. Number of scans of each sample are as follows: (a) COS: 7264, (b) COS-HCl: 640, (c) COS-Ace: 1192, (d) COS-Lat: 808, (e) COS-Cit: 8500, (f) COS-Tar: 5000, (g) COS-Mal: 744, (h) COS-Sal: 5000, (i) COS-Suc: 8500, (j) COS-Adi: 5000, and (k) COS-Glu: 1240.

X-ray diffraction (XRD) measurement of chitooligosaccharide and its salts was investigated with a D8 Advance diffractometer (Bruker, Billerica, MA, USA) with Cu target (λ = 0.154 nm) at 40 Kv, and the scanning scope of 2θ was 5–50°.

Differential scanning calorimetry (DSC) was performed using the Pyris Diamond DSC (PerkinElmer, Waltham, MA, USA). Chitooligosaccharide and its salts placed into an aluminum cup and sealed in an empty pan were heated from 5 to 300 °C at 10 °C/min. Nitrogen gas was used to confirm the thermal behavior.

Thermogravimetric (TGA) thermograms of chitooligosaccharide and its salts were measured using a thermogravimetric analyzer (TGA7, Perkin-Elmer, Waltham, MA, USA). Chitooligosaccharide and its salts of 5–10 mg were weighted and placed into an aluminum pan. The measurements were conducted from 5 to 300 °C at 10 °C/min under nitrogen purge.

### 3.3. Cell Culture

Murine macrophages RAW 264.7 were obtained from American Type Culture Collection (Manassas, VA, USA) and maintained in RPMI 1640 medium containing 10% fetal bovine serum (FBS), L-glutamine (2 mM), and 1% penicillin-streptomycin. The cells were cultured in an incubator at 37 °C and 5% CO_2_ concentration.

### 3.4. Cell Viability Assay

Cell viability of pure chitooligosaccharides (COS) and chitooligosaccharides with different salt types was evaluated by 3-(4,5-dimethyltiazol-2yl)-2,5-diphenyl-tetrazolium bromide (MTT) assay. Briefly, 100 μL RPMI complete medium containing 1 × 10^5^ RAW 264.7 cells (cells/mL) was placed in a 96-well flat-bottom culture plate. After incubation for 24 h, different concentrations of pure chitooligosaccharides and chitooligosaccharides with different salt types (0–500 μg/mL) were added, and then incubated for 24 h. Then, 100 μL MTT solution (0.5 mg/mL) was used to replace the culture medium. After 4 h, 100 μL MTT stopping buffer was added and cultured for 18 h, then the absorbance at 550 nm was measured with an Infinite M200 Pro spectrophotometer (Tecan, Männedorf, Switzerland).

### 3.5. Determination of Nitric Oxide (NO) Production in RAW 264.7 Cells

In order to quantify the production of NO, cells were seeded in a 96-well flat-bottom culture plate, and pre-cultured for 24 h, then pure chitooligosaccharides and chitooligosaccharides with different salt types (500 μg/mL) were added and cultured at 37 °C for 24 h, and the supernatant was collected and mixed with Griess reagent of the same volume (100 μL) and incubated for 10 min at room temperature. The absorbance at 570 nm was measured by an Infinite M200 Pro spectrophotometer, and the nitrite concentration was calculated according to the standard curve of sodium nitrite dilution.

### 3.6. Statistical Analysis

The data were expressed as mean ± standard deviation (SD). The statistical significance of the differences was evaluated by one-way analysis of variance (ANOVA) and Duncan’s multiple comparison test. *p*-value less than 0.05 was considered to be statistically significant.

## 4. Conclusions

Ten kinds of chitooligosaccharide salts, such as chitooligosaccharide hydrochloride, chitooligosaccharide lactate, chitooligosaccharide citrate, chitooligosaccharide glutamic acid salt, etc., were prepared. Their structures were analyzed by means of SEM, FTIR, NMR, and other modern analysis techniques. The characteristic peaks of each salt type of chitooligosaccharides corresponding to the acid type were determined. Their cytotoxicity and immune activity were studied. It was found that they had no cytotoxicity and could promote the growth of cells and could promote the secretion of NO by macrophages. Pure chitooligosaccharides had the best effect on promoting the growth and secretion of NO, which may be related to the high content of glucosamine at the same concentration, but the specific reasons need to be further studied.

## Figures and Tables

**Figure 1 molecules-26-02563-f001:**
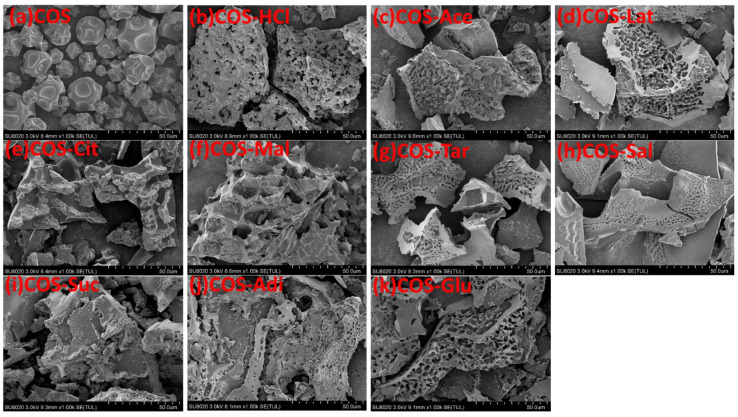
Scanning electron photomicrographs of: (**a**) Pure chitooligosaccharides (COS), (**b**) chitooligosaccharide hydrochloride (COS-HCl), (**c**) Chitooligosaccharide acetate (COS-Ace), (**d**) chitooligosaccharide lactate (COS-Lat), (**e**) chitooligosaccharide citrate (COS-Cit), (**f**) chitooligosaccharide malate (COS-Mal), (**g**) chitooligosaccharide tartrate (COS-Tar), (**h**) chitooligosaccharide acetylsalicylate (COS-Sal), (**i**) chitooligosaccharide succinate (COS-Suc), (**j**) chitooligosaccharide adipate (COS-Adi), and (**k**) chitooligosaccharide glutamate (COS-Glu).

**Figure 2 molecules-26-02563-f002:**
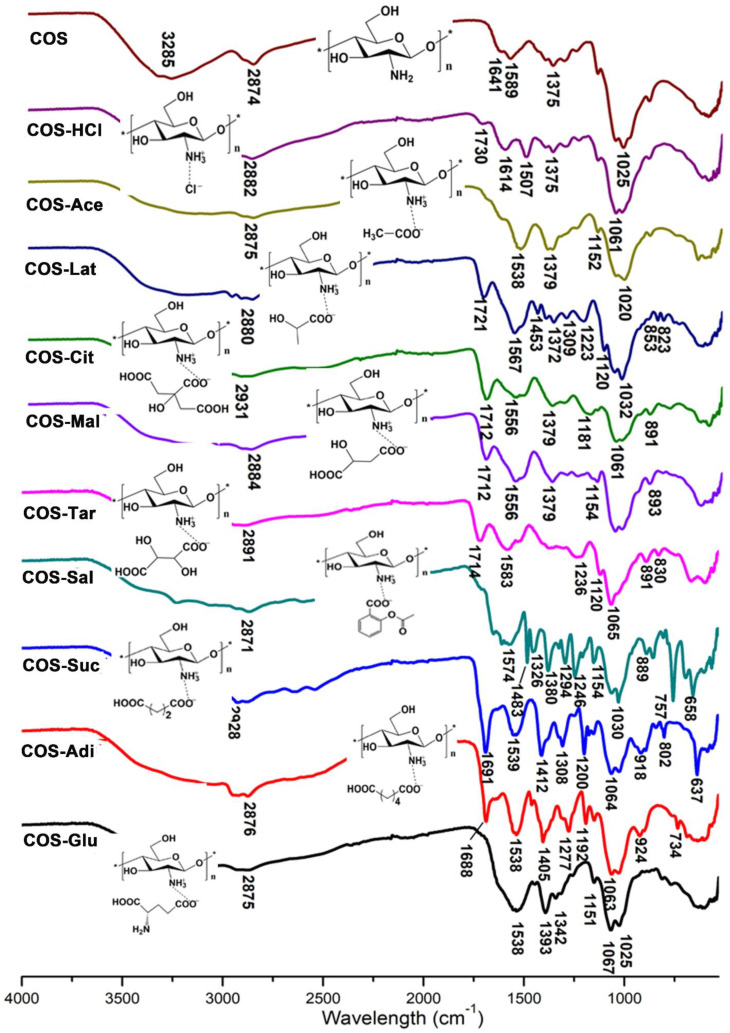
FTIR of chitooligosaccharide and its salts.

**Figure 3 molecules-26-02563-f003:**
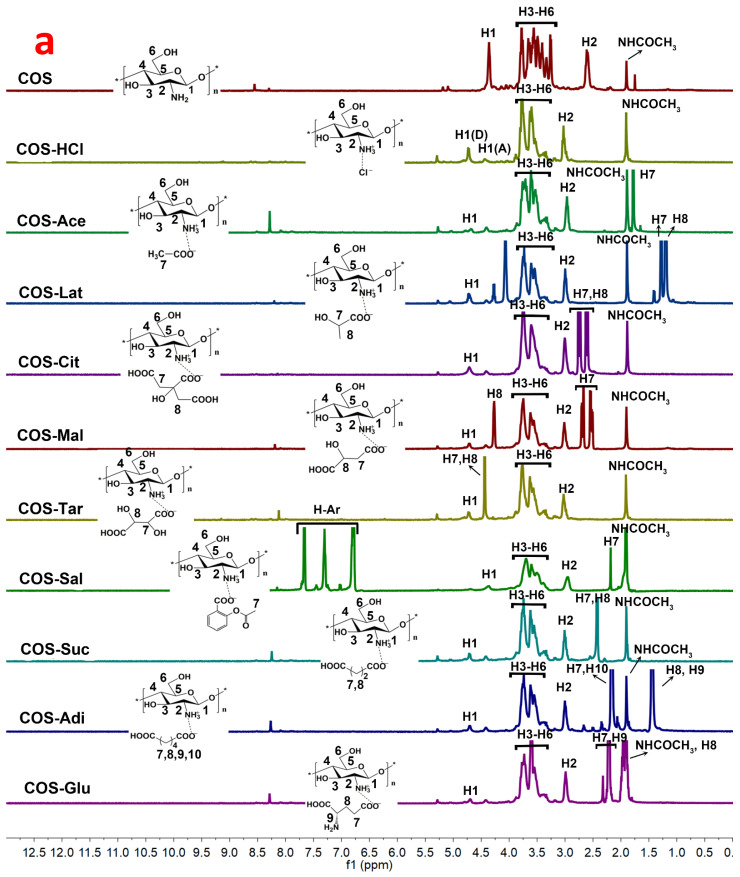
NMR spectroscopy of chitooligosaccharide and its salts. (**a**) ^1^HNMR and (**b**) ^13^CNMR.

**Figure 4 molecules-26-02563-f004:**
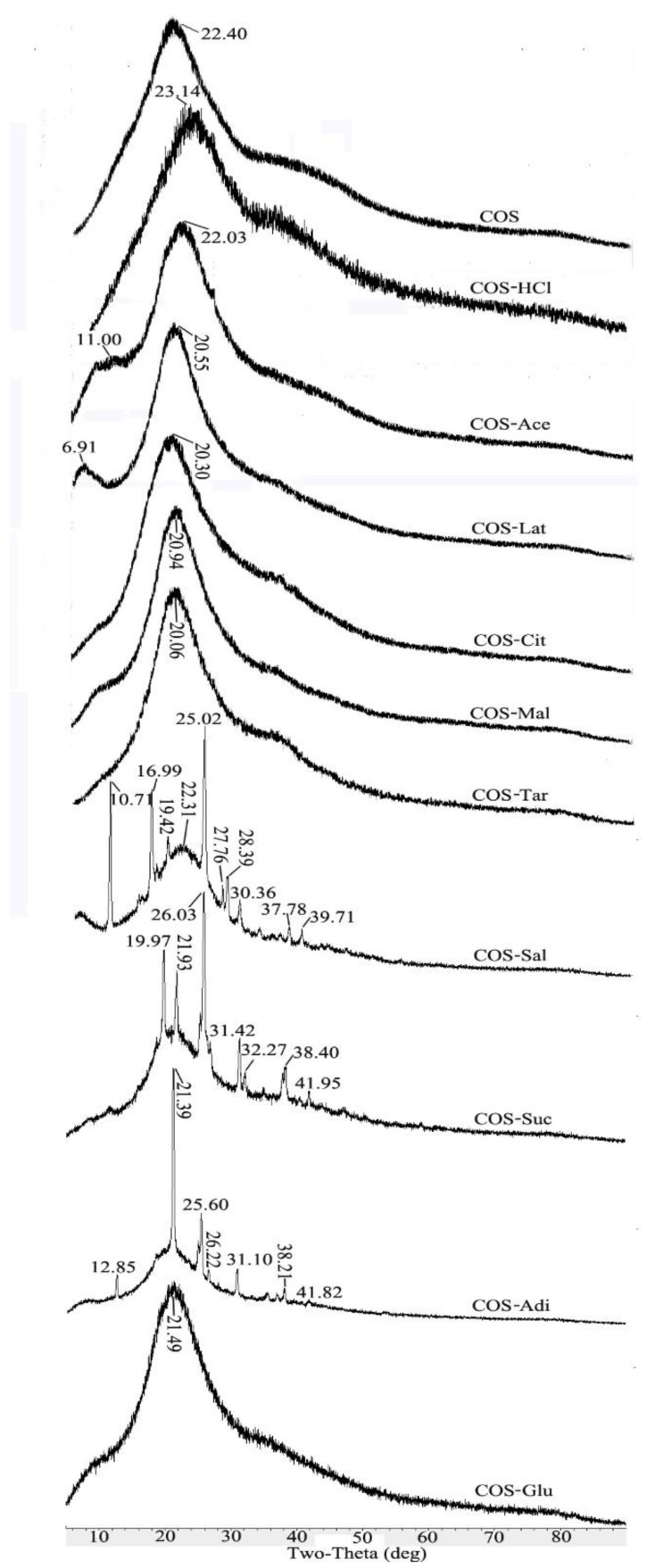
X-ray diffraction patterns of pure chitooligosaccharides (COS) and different salt type chitooligosaccharides.

**Figure 5 molecules-26-02563-f005:**
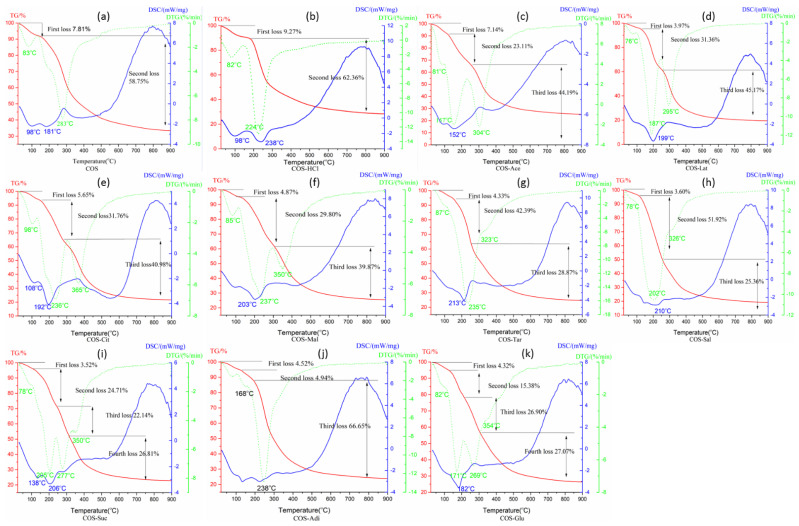
Thermal behavior of chitooligosaccharide and its salts. (**a**) Pure chitooligosaccharides (COS), (**b**) chitooligosaccharide hydrochloride (COS-HCl), (**c**) Chitooligosaccharide ace-tate (COS-Ace), (**d**) chitooligosaccharide lactate (COS-Lat), (**e**) chitooligosaccharide citrate (COS-Cit), (**f**) chitooligosac-charide malate (COS-Mal), (**g**) chitooligosaccharide tartrate (COS-Tar), (**h**) chitooligosaccharide acetylsalicylate (COS-Sal), (**i**) chitooligosaccharide succinate (COS-Suc), (**j**) chitooligosaccharide adipate (COS-Adi), and (**k**) chitooligo-saccharide glutamate (COS-Glu).

**Figure 6 molecules-26-02563-f006:**
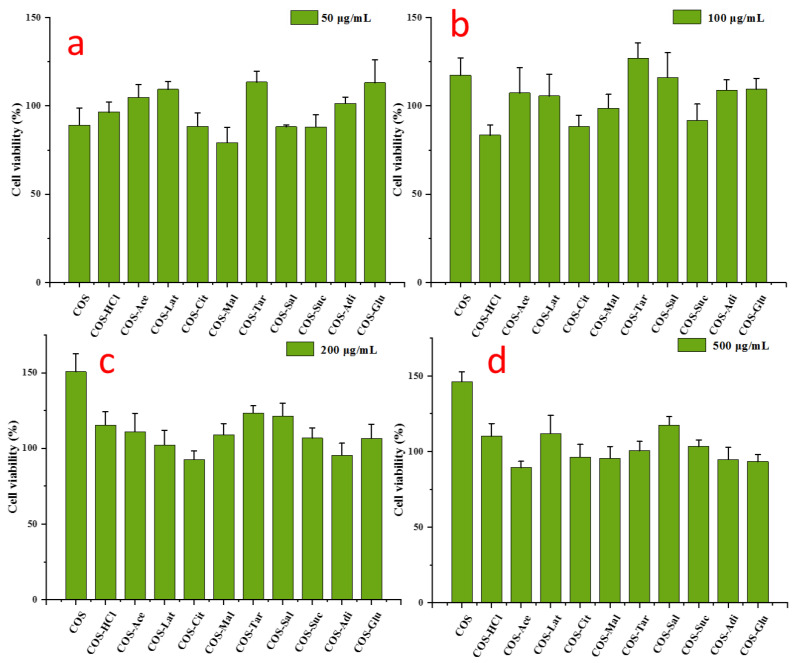
The effects of chitooligosaccharide and its salts on cell viability. RAW 264.7 cells were treated with chitooligosaccharide and its salts (50–500 μg/mL) for 24 h. (**a**) 50 μg/mL; (**b**) 100 μg/mL; (**c**) 200 μg/mL; (**d**) 500 μg/mL. The values are presented as the mean ± SD (n = 3, which refers to the number of replicates).

**Figure 7 molecules-26-02563-f007:**
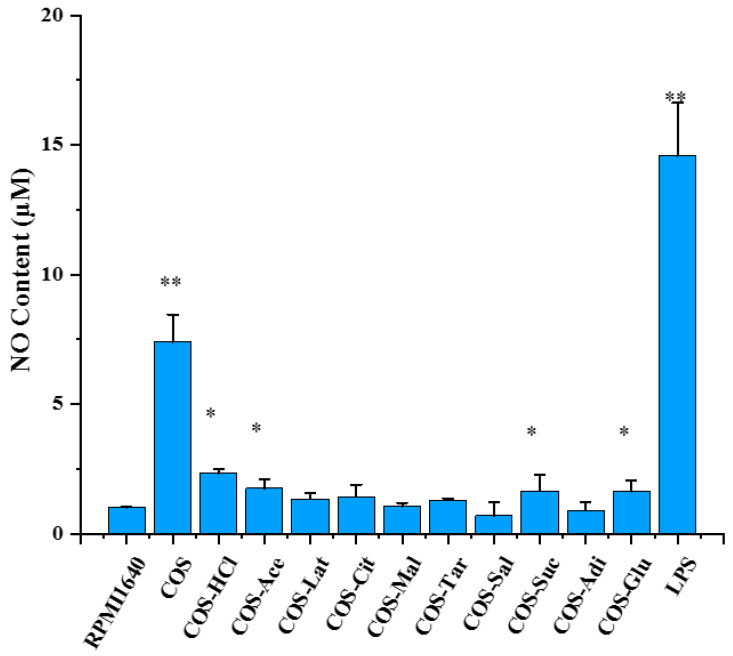
The effects of chitooligosaccharide and its salts on nitric oxide production. RAW 264.7 cells were treated with chitooligosaccharide and its salts (500 μg/mL) or lipopolysaccharide (1 μg/mL) for 24 h. The values are presented as the mean ± SD (n = 3). * *p* < 0.05, ** *p* < 0.01.

**Figure 8 molecules-26-02563-f008:**
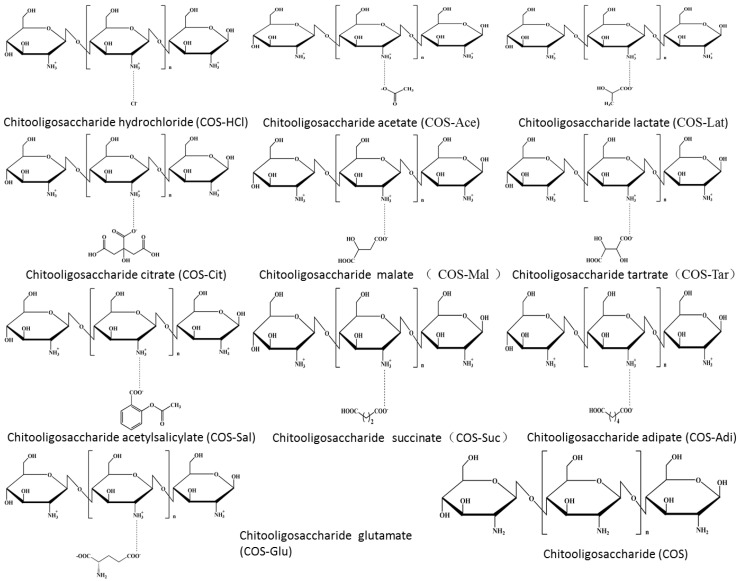
Chemical structure of chitooligosaccharide and its salts.

**Table 1 molecules-26-02563-t001:** Preparation conditions and product analysis of chitooligosaccharides with different salt forms.

Number	Content of Chitosan (g)	Acid Species	Acid Concentration	Yield (%)	Molecular Weight (Da)
1	5	acetic acid	2%	110	981
2	5	hydrochloric acid	2%	104	1230
3	5	citric acid	5%	182	3039
4	5	malic acid	3%	136	867
5	5	tartaric acid	4%	148	1780
6	5	adipic acid	2%	100	1089
7	5	succinic acid	2%	140	1030
8	5	acetylsalicylic acid	5%	162	1799
9	5	lactic acid	4%	114	1137
10	5	glutamic acid	3%	146	1020

## Data Availability

Not applicable.

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
