# Peer review of "Characterization of Different Salt Forms of Chitooligosaccharides and Their Effects on Nitric Oxide Secretion by Macrophages"

_molecules, 2021, doi:10.3390/molecules26092563_

Round 1

Reviewer 1 Report

Xing et al., have prepared various chitooligosaccharides and evaluated their physical and chemical characteristics in comparison to native form.

Title: ‘nitric oxide’ should be appropriate instead of ‘NO’

Line 33-36 should be rewritten. Flow is missing.

Line 79-80: The authors say all the salt forms exhibited honeycomb morphology. However, Cit, Mal, Suc forms look different pattern. Authors should change figures.

Figure 6 legend: “Mean ± SD (n=3)” refers to the number of experiments or replicates?

Line 296: Raw 264.7 ------- RAW 264.7

Figure 7: Statistical significance is missing. It looks like COS has significant NO production compared to Cells alone.

Evaluation of NO alone would not provide much information. It would be great if authors evaluate other parameters, like secreted cytokines or functional assays, to the effect of salt forms on macrophages.

Line 391-362: Is this procedure correct? please check. What is the necessity to incubate 18 h after blocking the reaction?

Line 398: It is strange, in the figure 7 legend, the authors mentioned 500 µg/ml, and in contrast, here it is 200 µg/ml. Which one is correct?

Author Response

Dear editor and reviewers:

Thanks for your time to review our revised manuscript (molecules-1202355). The manuscript has been carefully revised according to reviewers' comments. The following are the reviewer’s comments related to the manuscript and how we have addressed each of the reviewer’s concerns. Changes have been marked as red in the manuscript.

  1. Title: ‘nitric oxide’ should be appropriate instead of ‘NO’

Revised: In title, NO has been revised “nitric oxide”.

  1. Line 33-36 should be rewritten. Flow is missing.

Revised: Line 33-36 has been rewritten, the amendments are as follows:

In line 33-38, “solvent, only soluble in some organic acids and dilute inorganic acid such as acetic acid, dilute hydrochloric acid, etc. its application is greatly limited. Therefore, the improvement of the solubility of chitosan has been one of the concerns of researchers at home and abroad. Some studies have shown that chitosan salt formed with organic and inorganic acids is soluble in water and has been widely studied as a polymer drug carrier and a novel drug absorption accelerator across nasal and intestinal epithelium [17-18].”

  1. Line 79-80: The authors say all the salt forms exhibited honeycomb morphology. However, Cit, Mal, Suc forms look different pattern. Authors should change figures.

Revised: This part of the narrative has been modified. The revision is as follows:

“however, in addition to chitooligosaccharide citrate (COS•Cit), chitooligosaccharide malate (COS•Mal) and chitooligosaccharide succinate (COS•Suc), other different salt products are massive with different honeycomb sizes. COS•Cit, COS•Mal and COS•Suc are all massive, so their structures are similar. For other salt types, although the size of the honeycomb is different,”

  1. Figure 6 legend: “Mean ± SD (n=3)” refers to the number of experiments or replicates?

Revised: “Mean ± SD (n=3)” refers to the number of replicates.

  1. Line 296: Raw 264.7 ------- RAW 264.7

Revised: “Raw 264.7” has been revised “RAW 264.7”.

  1. Figure 7: Statistical significance is missing. It looks like COS has significant NO production compared to Cells alone.

Revised: Figure 7 shows significant differences. Figure 7 has been revised in this paper.

  1. Evaluation of NO alone would not provide much information. It would be great if authors evaluate other parameters, like secreted cytokines or functional assays, to the effect of salt forms on macrophages.

Answered: Thank you very much for your valuable comments. We will further improve the other parameters in immunoassay, such as the determination of cytokine secretion level, cell uptake measurement, etc.

  1. Line 391-362: Is this procedure correct? please check. What is the necessity to incubate 18 h after blocking the reaction?

Answered: Thank you for your careful checking. MTT is a method for detecting cell survival and growth. The detection principle is that the succinate dehydrogenase in the mitochondria of living cells can reduce the exogenous MTT to water-insoluble blue-purple crystal formazan and deposit it in the cells, while dead cells have no such function. MTT stop solution (10% SDS-0.01M HCl) can dissolve formazan in cells. Because formazan is difficult to dissolve after crystallization, it is necessary to react with MTT stop solution at 37°C for 18 h to ensure that the crystals can be completely dissolved and the accuracy of the results. Then use the enzyme-linked immunoassay to measure its light absorption value, which can indirectly reflect the number of living cells.

  1. Line 398: It is strange, in the figure 7 legend, the authors mentioned 500 µg/ml, and in contrast, here it is 200 µg/ml. Which one is correct?

Answered: Sorry, we wrote it wrong. It should be 500µg/ml.

Reviewer 2 Report

The Authors surely put a lot of work in the synthesis and analysis of the studied materials (different salts of chitooligosaccharides. However, there are some major issued related to this manuscript that are listed below. What is most important, the English must be improved and in the introduction the previous studies on those salts must be cited.

  1. Lines 33-36, this sentence is very long, please rewrite.
  2. Lines 62-65, it was already mentioned in lines 32-35, please remove.
  3. “Rarely reported”-does it mean that the salts that you have prepared have been obtained before? This must be clarified.
  4. In the introduction it must be stated clearly which of the salts discussed in this study have been obtained before. Is there at least one novel salt? Further, it should be stated if those salts have been studied using methods used by the Authors (SEM, FT-IR, NMR). The results obtained by the Authors and the results reported previously must be compared and carefully discussed.
  5. The Authors use term “chromatogram” (line 103) instead of “spectrum”. This must be corrected.
  6. Line 106, why capital “C”?
  7. Line 146, it should be “13C”, please correct this caption.
  8. Figure 3, spectrum of COS, what are those two peaks (next to C4 and C5)? Why they are not assigned?
  9. Lines 225-232, the PXRD of pure Sal, Suc and Adi should be compared with those of COS salts in order to deny the possibility that some of the acids did not react fully and have recrystallized forming a mixture (two phases). If it is not possible to measure those PXRD experimentally than simulate them using available crystal structures.
  10. Line 294, it is not fully described why this particular biological test has been performed? In the introduction the immunomodulatory effect of COS is not mentioned.
  11. Line 334, it should be “Methods”
  12. Lines 365-366, this should be rewritten. Instead of “performed” use “recorded” and please delete “by nuclear magnetic resonance”.
  13. The method of NMR analysis is not described in a proper way. There are not enough details, i.e. on pulses, acquisition parameters etc. Please consult this part with a specialist.
  14. Line 372, what was the mass of the sample?
  15. Line 412, what is a NMC? It should be NMR.
  16. The cited literature is old, most of the cited works were published before 2010. Please update.

Author Response

Dear editor and reviewers:

Thanks for your time to review our revised manuscript (molecules-1202355). The manuscript has been carefully revised according to reviewers' comments. The following are the reviewer’s comments related to the manuscript and how we have addressed each of the reviewer’s concerns. Changes have been marked as red in the manuscript.

  1. Lines 33-36, this sentence is very long, please rewrite.

Revised: Line 33-36 has been rewritten, the amendments are as follows:

In line 33-38, “solvent, only soluble in some organic acids and dilute inorganic acid such as acetic acid, dilute hydrochloric acid, etc. its application is greatly limited. Therefore, the improvement of the solubility of chitosan has been one of the concerns of researchers at home and abroad. Some studies have shown that chitosan salt formed with organic and inorganic acids is soluble in water and has been widely studied as a polymer drug carrier and a novel drug absorption accelerator across nasal and intestinal epithelium [17-18].”

  1. Lines 62-65, it was already mentioned in lines 32-35, please remove.

Revised: the sentence “can be dissolved in many organic acids and dilute inorganic acids, such as acetic acid, lactic acid, citric acid, tartaric acid, glutamic acid, succinic acid, hydrochloric acid and so on. Chitosan” has been remove.

  1. “Rarely reported”-does it mean that the salts that you have prepared have been obtained before? This must be clarified.

Revised: “Rarely reported” has been revised “have not been reported.” In fact, chitosan citrate and chitosan glutamate have been reported. However, the chitosan was dissolved in the corresponding acid and spray dried to prepare the corresponding salt, not the corresponding chitooligosaccharide salt.

  1. In the introduction it must be stated clearly which of the salts discussed in this study have been obtained before. Is there at least one novel salt? Further, it should be stated if those salts have been studied using methods used by the Authors (SEM, FT-IR, NMR). The results obtained by the Authors and the results reported previously must be compared and carefully discussed.

Revised: We have added research results of some authors.

“For example, Hai et al. [16] dissolved chitosan in lactic acid and degraded chitosan with hydrogen peroxide to obtain chitooligosaccharide lactate. 1HNMR showed the product of chitooligosaccharide lactate prepared by them has one more hydrogen absorption peak in methyl group than chitosan. Li et al. [17] dissolved chitosan in glacial acetic acid and prepared chitooligosaccharide acetate by enzymatic hydrolysis of chitosan. 13CNMR showed that a new peak appeared at about 22ppm, which could be explained as the absorption peak of methyl group in chitooligosaccharide acetate. Qin et al. [18] dissolved chitosan in hydrochloric acid, and obtained chitooligosaccharide hydrochloride by oxidative degradation of chitosan. According to 13CNMR, its structure is consistent with that of chitosan. Some studies showed the potential of COS as immunostimulatory agent. Zhang et al. [19] showed that chitooligosaccharide exerts this immunostimulating activity by interacting with membrane receptors on the macrophage surface. Zheng et al. [20] found that chitooligosaccharide has promoted the expression of the genes of vital molecules in NF-κB and AP-1 pathways and induced the phosphorylation of protein in RAW 264.7 macrophage. All the raw materials they used were chitooligosaccharide acetate. In addition to chitooligosaccharide acetate, chitooligosaccharide hydrochloride and chitooligosaccharide lactate, other salt form chitooligosaccharides have not been reported.”

  1. The Authors use term “chromatogram” (line 103) instead of “spectrum”. This must be corrected.

Revised: “chromatogram” has been revised “spectrum”

  1. Line 106, why capital “C”?

Revised: Sorry, we wrote it wrong. Capital “C” has been changed to lowercase “c”.

  1. Line 146, it should be “13C”, please correct this caption.

Answered: It has been modified accordingly. In this paper, 1HNMR and 13CNMR are illustrated together in Figure 3.

  1. Figure 3, spectrum of COS, what are those two peaks (next to C4 and C5)? Why they are not assigned?

Answered: It can be seen from the 13CNMR of cos that there are acetylamino groups in COS. We speculate that the peaks next to C4 and C5 may be the peaks of C4 and C5 on the acetylglucosamine unit.

  1. Lines 225-232, the PXRD of pure Sal, Suc and Adi should be compared with those of COS salts in order to deny the possibility that some of the acids did not react fully and have recrystallized forming a mixture (two phases). If it is not possible to measure those PXRD experimentally than simulate them using available crystal structures.

Answered: In this paper, we mainly want to find out whether the structural differences of ten different salt chitooligosaccharides are related to the introduction of acids, which is partly proved by references. In the future, we will carry out specific structure characterization and activity determination for some active salt types, so that chitooligosaccharides can be better applied.

  1. Line 294, it is not fully described why this particular biological test has been performed? In the introduction the immunomodulatory effect of COS is not mentioned.

Revised: In introduction, I have added the immunomodulatory effect of COS.

“For example, Hai et al. [16] dissolved chitosan in lactic acid and degraded chitosan with hydrogen peroxide to obtain chitooligosaccharide lactate. 1HNMR showed the product of chitooligosaccharide lactate prepared by them has one more hydrogen absorption peak in methyl group than chitosan. Li et al. [17] dissolved chitosan in glacial acetic acid and prepared chitooligosaccharide acetate by enzymatic hydrolysis of chitosan. 13CNMR showed that a new peak appeared at about 22ppm, which could be explained as the absorption peak of methyl group in chitooligosaccharide acetate. Qin et al. [18] dissolved chitosan in hydrochloric acid, and obtained chitooligosaccharide hydrochloride by oxidative degradation of chitosan. According to 13CNMR, its structure is consistent with that of chitosan. Some studies showed the potential of COS as immunostimulatory agent. Zhang et al. [19] showed that chitooligosaccharide exerts this immunostimulating activity by interacting with membrane receptors on the macrophage surface. Zheng et al. [20] found that chitooligosaccharide has promoted the expression of the genes of vital molecules in NF-κB and AP-1 pathways and induced the phosphorylation of protein in RAW 264.7 macrophage. All the raw materials they used were chitooligosaccharide acetate. In addition to chitooligosaccharide acetate, chitooligosaccharide hydrochloride and chitooligosaccharide lactate, other salt form chitooligosaccharides have not been reported.”

  1. Line 334, it should be “Methods”

Revised: “methods” has been revised as “Methods”.

  1. Lines 365-366, this should be rewritten. Instead of “performed” use “recorded” and please delete “by nuclear magnetic resonance”.

Revised: This part of content has been rewritten.

1HNMR and 13CNMR spectra were recorded on a 500 MHz Agilent DD2 500 spectrometer for structural analysis in chitooligosaccharide and its salts. Measurements were recorded at 25℃. Samples were dissolved at D2O (10 mg/mL) and pulse sequence was s2pul, relaxation delay was 1.0000s. Acquisition time was 2.0447 s with 16 scans and spectral width was 8012.8 Hz for 1HNMR. Acquisition time was 1.0486 s and spectral width was 31250 Hz for 13CNMR. For 13CNMR, number of scans of different samples are different. Number of scans of each sample are as follows: (a) COS: 7264, (b) COS•HCl:640, (c) COS•Ace: 1192, (d) COS•Lat: 808, (e) COS•Cit: 8500, (f) COS•Tar: 5000, (g) COS•Mal: 744, (h) COS•Sal: 5000, (i) COS•Suc: 8500, (j) COS•Adi: 5000, and (k) COS•Glu: 1240.”

  1. The method of NMR analysis is not described in a proper way. There are not enough details, i.e. on pulses, acquisition parameters etc. Please consult this part with a specialist.

Revised: Detail parameters have been added.

1HNMR and 13CNMR spectra were recorded on a 500 MHz Agilent DD2 500 spectrometer for structural analysis in chitooligosaccharide and its salts. Measurements were recorded at 25℃. Samples were dissolved at D2O (10 mg/mL) and pulse sequence was s2pul, relaxation delay was 1.0000s. Acquisition time was 2.0447 s with 16 scans and spectral width was 8012.8 Hz for 1HNMR. Acquisition time was 1.0486 s and spectral width was 31250 Hz for 13CNMR. For 13CNMR, number of scans of different samples are different. Number of scans of each sample are as follows: (a) COS: 7264, (b) COS•HCl:640, (c) COS•Ace: 1192, (d) COS•Lat: 808, (e) COS•Cit: 8500, (f) COS•Tar: 5000, (g) COS•Mal: 744, (h) COS•Sal: 5000, (i) COS•Suc: 8500, (j) COS•Adi: 5000, and (k) COS•Glu: 1240.”

  1. Line 372, what was the mass of the sample?

Revised: The mass of the sample is 10 mg/mL.

  1. Line 412, what is a NMC? It should be NMR.

Revised: Sorry, we wrote it wrong. “NMC” has been changed to “NMR”.

  1. The cited literature is old, most of the cited works were published before 2010. Please update.

Revised: The cited literature has been updated.

Round 2

Reviewer 1 Report

Yes, the authors have answered the reviewer(s) comments, appropriately. 

I hereby endorse the manuscript for the publication. 

Reviewer 2 Report

The authors have answered my comments and made changes in their manuscript and so I can suggest the manuscript for the publication.